# The mutational signatures of poor treatment outcomes on the drug-susceptible *Mycobacterium tuberculosis* genome

Yiwang Chen[1,2], Qi Jiang[3], Mijiti Peierdun[4], Howard E Takiff[5], Qian Gao[1,2]*

[1]Key Laboratory of Medical Molecular Virology (MOE/NHC/CAMS), School of Basic Medical Sciences, Shanghai Medical College, Shanghai Institute of Infectious Disease and Biosecurity, Fudan University, Shanghai, China; [2]National Clinical Research Center for Infectious Diseases, Shenzhen Third People's Hospital, Shenzhen, China; [3]School of Public Health, Public Health Research Institute of Renmin Hospital, Wuhan University, Wuhan, China; [4]Department of Epidemiology and Biostatistics,School of Public Health, Xinjiang Medical University, Urumqi, China; [5]Instituto Venezolano de Investigaciones Cientificas (IVIC), Caracas, Venezuela

*For correspondence:
qiangao@fudan.edu.cn

**Abstract** Drug resistance is a known risk factor for poor tuberculosis (TB) treatment outcomes, but the contribution of other bacterial factors to poor outcomes in drug-susceptible TB is less well understood. Here, we generate a population-based dataset of drug-susceptible *Mycobacterium tuberculosis* (MTB) isolates from China to identify factors associated with poor treatment outcomes. We analyzed whole-genome sequencing (WGS) data of MTB strains from 3196 patients, including 3105 patients with good and 91 patients with poor treatment outcomes, and linked genomes to patient epidemiological data. A genome-wide association study (GWAS) was performed to identify bacterial genomic variants associated with poor outcomes. Risk factors identified by logistic regression analysis were used in clinical models to predict treatment outcomes. GWAS identified fourteen MTB fixed mutations associated with poor treatment outcomes, but only 24.2% (22/91) of strains from patients with poor outcomes carried at least one of these mutations. Isolates from patients with poor outcomes showed a higher ratio of reactive oxygen species (ROS)-associated mutations compared to isolates from patients with good outcomes (26.3% vs 22.9%, t-test, p=0.027). Patient age, sex, and duration of diagnostic delay were also independently associated with poor outcomes. Bacterial factors alone had poor power to predict poor outcomes with an AUC of 0.58. The AUC with host factors alone was 0.70, but increased significantly to 0.74 (DeLong's test, p=0.01) when bacterial factors were also included. In conclusion, although we identified MTB genomic mutations that are significantly associated with poor treatment outcomes in drug-susceptible TB cases, their effects appear to be limited.

## Editor's evaluation

In this useful study, a Genome Wide Association-type analysis is applied to clinical *Mycobacterium tuberculosis* isolates to discover genetic polymorphisms linked to poor tuberculosis outcomes. The evidence for the detected associations is still incomplete, as the corresponding polymorphisms are not adequate to power a prediction model for infection outcome, although key host factors – including patient age, sex, and duration of diagnostic delay (which have stronger predictive value) – appear to enhance predictive capacity. The work will be of interest to clinical microbiologists.

## Introduction

Tuberculosis (TB), caused by *Mycobacterium tuberculosis* (MTB), is responsible for more deaths globally than any other single infectious agent, causing nearly 1.5 million deaths and an estimated 10 million new cases each year (*World Health Organization, 2022*). Successful treatment of TB not only cures the patient but also prevents disease transmission and the development of difficult-to-treat drug-resistant strains. Treatment outcomes are therefore important metrics for assessing the effectiveness of national TB control programs (*Dheda et al., 2017*; *Migliori et al., 2019*). While approximately 86% of patients with drug-susceptible TB are cured with the standard four-drug treatment (*World Health Organization, 2022*), there remains a substantial subset of patients who fail treatment. To formulate strategies that will reduce treatment failures, it would be helpful to first define the risk factors associated with poor outcomes.

There are host factors that are well known to be associated with treatment failure, including poor patient adherence (*Alipanah et al., 2018*), sex, age (*Imperial et al., 2018*), diagnostic delay (*Lestari et al., 2020*), co-infection with the human immunodeficiency virus (HIV) and TB treatment history (*Bastos et al., 2017*; *Chenciner et al., 2021*), but we wondered whether the genetic composition of the infecting strain might also contribute to poor outcomes. While drug resistance is the major bacterial risk factor for TB treatment failure (*Lange et al., 2019*; *Mirzayev et al., 2021*), a growing number of genomic studies suggest that other bacterial determinants may also be risk factors. For example, specific mutations in metabolism-related genes of MTB could lead to drug tolerance (*Hicks et al., 2018*; *Torrey et al., 2016*), thereby increasing the risk of developing resistance and relapsing after treatment (*Brauner et al., 2016*; *Liu et al., 2020a*). (*Liu et al., 2022*) recently showed that genetic mutations in the gene encoding the transcriptional regulator *resR* can cause antibiotic resilience and are associated with the acquisition of drug resistance and treatment failure. Although these genetic polymorphisms in the MTB genome do not directly lead to drug resistance, they are significantly more common in drug-resistant bacteria (*Hicks et al., 2018*; *Liu et al., 2022*). Because drug resistance is such a dominant risk factor for treatment failure, a search for other bacterial genomic determinants associated with treatment failure is best performed in drug-susceptible MTB isolates.

To study the role of bacterial genomic determinants, other than drug resistance mutations, that might be associated with poor treatment outcomes, we analyzed drug-susceptible TB isolates from new TB cases collected in population-based cohort studies at three different sites in China. The bacterial determinants we identified and the patient characteristics were then used to build a clinical prediction model to estimate the contribution of bacterial factors to poor TB treatment outcomes.

## Results

### Characteristics of the study population and MTB isolates

The pooled study population from the three different sites in China consisted of 3496 new cases of drug-susceptible TB. The patients were divided into three groups based on their treatment outcomes: good outcomes (88.8%, 3105/3496), poor outcomes (2.6%, 91/3496), and other outcomes (8.6%, 300/3496). To explore the bacterial factors associated with poor TB treatment outcomes, we first excluded patients with outcomes unlikely to be associated with bacterial factors, including patients lost to follow-up, non-TB deaths, and unknown outcomes. Ultimately, a total of 3196 new cases with drug-susceptible TB were included in the study (*Figure 1A*): 3105 with good outcomes and 91 with poor outcomes (failure, 25; TB death, 15; transferred for MDR, 4; and relapse, 47). The study patients were recruited from Shanghai (49.1%, 1569/3196), Sichuan (30.6%, 979/3196), and Heilongjiang (20.3%, 648/3196) provinces, China (*Figure 1A*). They had a mean age of 42.1 ± 18.2 years and 72.4% (2313/3196) were male. WGS was performed on all 3196 isolates, with an average depth of 100×and average genome coverage of 98%. Phylogenetic analysis of WGS data showed that nearly three-quarters of the isolates were lineage 2 (74.2%, 2373/3196), with more than half belonging to the modern Beijing sublineage L2.3 (54.9%, 1754/3196) (*Figure 1B*).

### Identification of a functional mutation set for predicting treatment outcomes

GWAS of the MTB isolates identified fourteen fixed nonsynonymous variants associated with poor treatment outcomes (*Figure 2A*). These variants were distributed in thirteen genes involved in

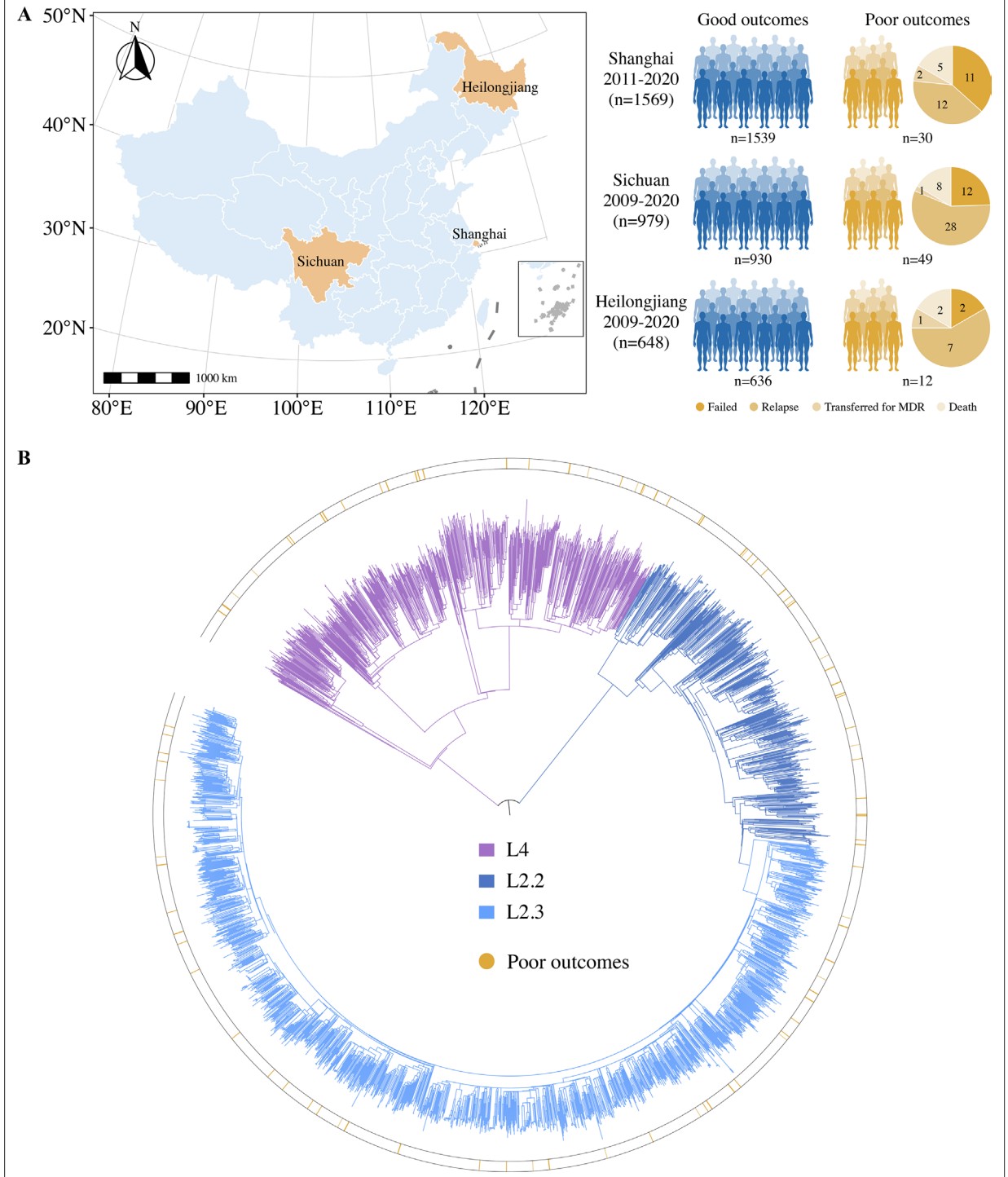

**Figure 1.** Sample origin and genetic structure of *Mycobacterium tuberculosis*. (**A**) Geographic location of the samples analyzed and study cohort characteristics. (**B**) The phylogenetic tree of 3196 drug-susceptible tuberculosis strains. The different colors on the branches indicate different lineages and sublineages. The outside circle indicates the treatment outcomes of corresponding patients.

intermediary metabolism, and respiration (*cobN*, *dlaT*, *metA*, *Rv0648,* and *Rv1248c*), cell wall and cell processes (*ctpB*, *Rv2164c,* and *Rv1717*) and virulence (*otsB1* and *Rv3168*), with the *otsB1* G559D mutation showing the strongest association (p=7.3 × 10$^{-10}$) (*Figure 2—source data 1*).

Unfixed mutations are thought to represent adaptive mutations emerging within the host (*Nimmo et al., 2020*). To investigate whether unfixed mutations affect treatment outcomes, we performed a

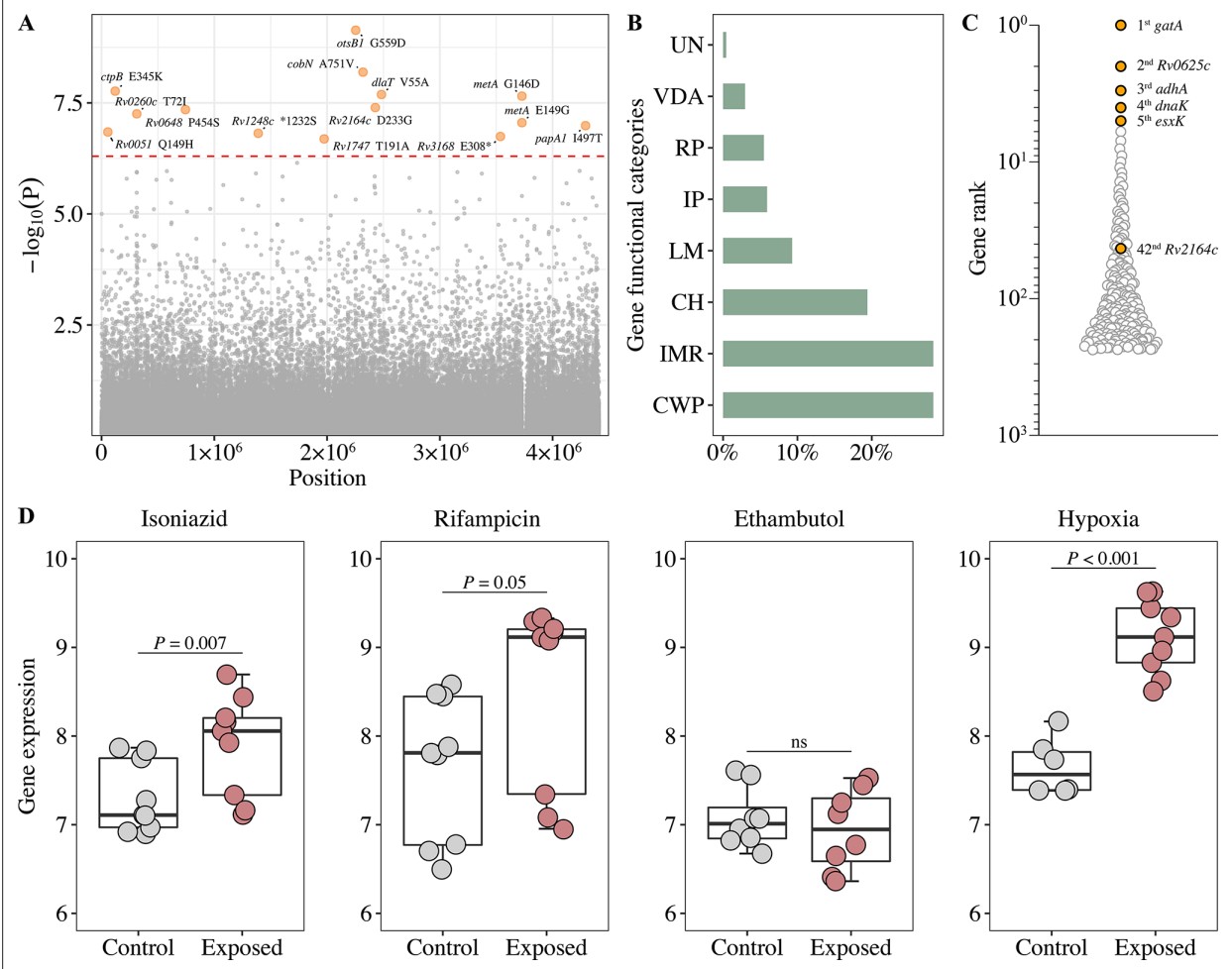

**Figure 2.** Generation of the functional mutation set. (**A**) Manhattan plots of genome-wide association study (GWAS) for fixed single nucleotide polymorphisms (SNPs) associated with poor treatment outcomes. The dashed red line highlights the Bonferroni-corrected threshold (p=5.04 × 10⁻⁷). (**B**) Distribution of GWAS identified unfixed SNPs across gene functional categories. CWP, cell wall, and cell processes; IMR, intermediary metabolism, and respiration; CH, conserved hypotheticals; LM, lipid metabolism; IP, information pathways; RP, regulatory proteins; VDA, virulence, detoxification, adaptation; UN, unknown. (**C**) Gene prioritization strategies (based on p-value rank) for significantly associated unfixed SNPs. (**D**) Gene expression from RNA-seq (log₂FPKM) of *Rv2164c* under drug pressure and hypoxia.

The online version of this article includes the following source data and figure supplement(s) for figure 2:

**Source data 1.** GWAS identified fixed SNPs.

**Figure supplement 1.** Manhattan plots of unfixed single nucleotide polymorphisms (SNPs) associated with poor treatment outcomes.

**Figure supplement 2.** Gene expression (log₂FPKM) from RNA-seq after drug exposure and hypoxia.

**Figure supplement 3.** Within-host frequency distribution of genome-wide association study (GWAS)-identified unfixed mutations.

**Figure supplement 4.** Manhattan plot of genome-wide association study (GWAS) analysis based on the Malawi dataset.

GWAS analysis of unfixed mutations and found that 237 mutations were associated with poor treatment outcomes (*Figure 2—figure supplement 1*). The frequency of these mutations was mostly in the range of 5–10%, and they were predominantly found in genes whose encoded proteins are involved in cell wall and cell processes, intermediary metabolism, and respiration (*Figure 2B*). When the genes carrying unfixed mutations were ranked according to the significance of their associations with poor outcomes, the highest ranked gene was *gatA*, which has been previously associated with rifampicin tolerance (*Cai et al., 2020*; *Figure 2C*).

Gene expression patterns under stress conditions can provide important insights into the gene's function (*Bosch et al., 2021*). We, therefore, analyzed the genes containing GWAS-identified fixed mutations for changes in expression after exposure to first-line drugs and hypoxic conditions. The

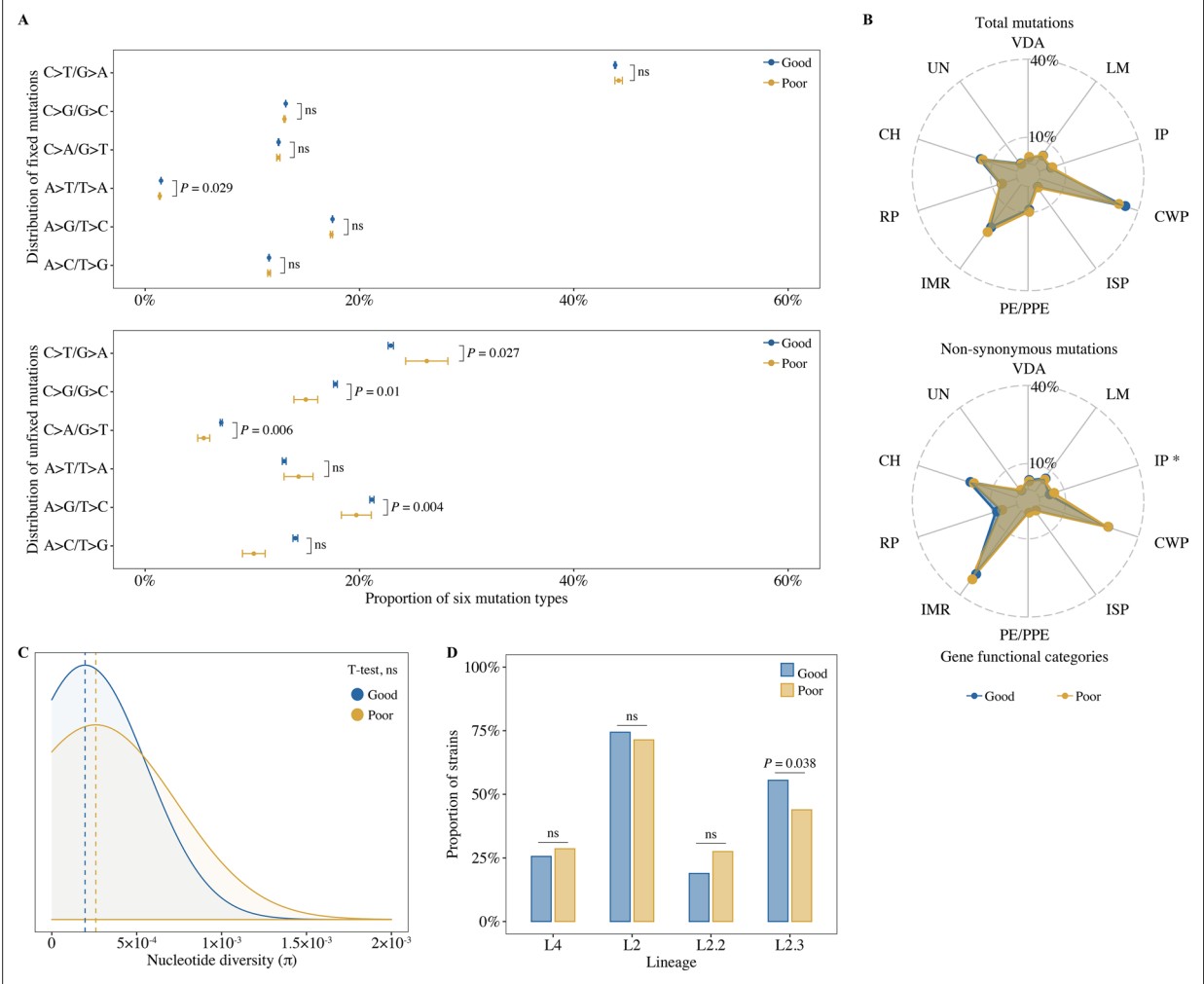

**Figure 3.** Bacterial whole-genome mutation features between patients with different treatment outcomes. (**A**) The proportion of six mutation types in all fixed and unfixed mutations (t-test, mean range: mean ± SE). (**B**) Distribution of total unfixed mutations and nonsynonymous unfixed mutations across gene functional categories (t-test). VDA, virulence, detoxification, adaptation; LM, lipid metabolism; IP, information pathways; CWP, cell wall, and cell processes; ISP, insertion seqs and phages; IMR, intermediary metabolism, and respiration; RP, regulatory proteins; CH, conserved hypotheticals; UN, unknown. (**C**) Comparison of nucleotide genetic diversity between isolated patients with good and poor outcomes (t-test). (**D**) Distribution of *Mycobacterium tuberculosi* (MTB) lineages and sublineages (chi-square test). p-value <0.05 was considered significant. *, p<0.05, ns, no significant.

expression of some of the genes increased under these conditions (drug-treated: *Rv2164c*, *cobN*, *Rv0260c*; hypoxia: *Rv2164c*, *otsB1*, *papA1*), but only *Rv2164c* (*Figure 2D*) contained both GWAS-identified fixed (*Figure 2A*) and unfixed (*Figure 2C*) mutations. The expression of other genes decreased (drug-treated: *otsB1*, *Rv0648*, *Rv1248c*, *Rv3168*; hypoxia: *Rv0648*) (*Figure 2D*; *Figure 2—figure supplement 2*), suggesting these genes could be involved in adaptation to drug and hypoxic stress.

## Ongoing mutational signatures of ROS associated with TB treatment outcomes

Previous reports have suggested that poor TB treatment outcomes may be associated with increased ROS mutational signatures (C>T/G>A mutations) (*Liu et al., 2020b*; *Moreno-Molina et al., 2021*). To determine whether the increased ROS signatures were the result of mutations that were fixed before the infection or were unfixed because they arose de novo during infection, we compared the distribution of six mutation types in fixed and unfixed mutations. In the fixed mutations, there was no significant difference in the proportions of ROS mutational signatures for isolates from patients with good or poor outcomes (43.9% vs 44.2%, t-test, p=0.364, *Figure 3A*). In the unfixed de novo mutations,

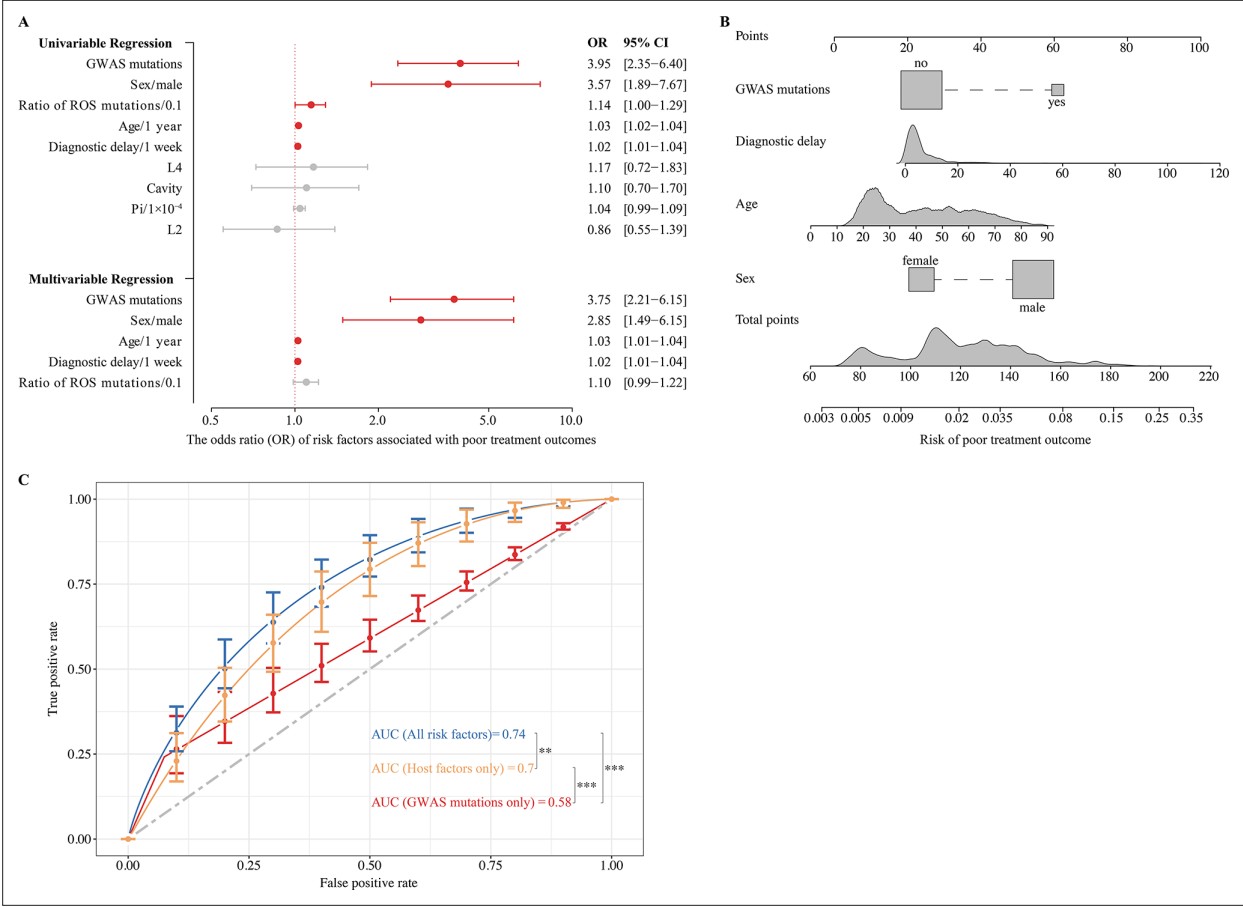

**Figure 4.** Effects of genome-wide association study (GWAS) identified mutations on tuberculosis treatment outcomes. (**A**) Univariable and multivariable logistic regression on the risk factors for poor treatment outcomes. (**B**) Nomogram for predicting the probability of poor treatment outcomes. (**C**) ROC curves are based on risk factors that may be predictive of tuberculosis treatment outcomes. p-value <0.05 was considered significant. *p<0.05, **p<0.01, ***p<0.001.

however, the ROS mutational signatures were significantly more frequent in isolates from patients with poor outcomes (26.3% vs 22.9%, t-test, p=0.027, **Figure 3A**). A further analysis of the distribution of unfixed mutations across gene functional categories found no difference in the distribution of total unfixed mutations between good and poor outcomes (**Figure 3B**), but isolates from patients with poor outcomes showed a higher percentage of nonsynonymous mutations (5.7% vs 3.9%, t-test, p=0.048, **Figure 3B**) in genes belonging to the functional category 'information pathway'.

Nucleotide diversity and the characteristics of the different lineages of MTB are thought to be determinants of virulence and thus may affect TB treatment outcomes. (**O'Neill et al., 2015**; **Tong et al., 2022**). An analysis of nucleotide diversity revealed no significant differences between isolates from patients with good or poor outcomes (2.0 × 10$^{-4}$ vs 2.6 × 10$^{-4}$, t-test, p=0.156, **Figure 3C**). In contrast, an analysis of lineage distribution showed that strains belonging to the modern Beijing lineage L2.3 were significantly more prevalent in patients with good outcomes (55.5% vs 44.0%, chi-square test, p=0.038, **Figure 3D**). Although lineage L2.3 has been associated with high virulence and increased transmission (**Tong et al., 2022**), lineage was not associated with poor treatment outcomes in the populations studied.

## GWAS-identified mutations help predict TB treatment outcomes

To identify the risk factors for poor TB treatment outcomes, we used logistic regression that included both the patients' clinical characteristics and the bacterial factors associated with poor outcomes. We found that patient age, sex, duration of diagnostic delay, and the GWAS-identified fixed mutations were all independently associated with poor TB outcomes (**Figure 4A**). We then performed logistic

regression with the identified risk factors to construct a clinical prediction model that is depicted graphically by the nomogram (*Figure 4B*). For example, in a 65-year-old male TB patient with a 4 month delay in diagnosis, the risk of poor outcome would increase from 5.4 to 17.4% if his MTB isolate contained at least one GWAS-identified mutation.

The ROC curve for the clinical prediction model based only on host risk factors had an AUC of 0.70, which was significantly higher than the AUC of 0.58 (DeLong's test, p<0.001) when only GWAS-identified bacterial mutations were included (*Figure 4C*). However, when both host factors and GWAS-identified mutations were included in the model, the AUC for predicting poor treatment outcomes increased significantly (DeLong's test, p=0.01) from 0.70 to 0.74, with a sensitivity of 0.85 and specificity of 0.55 (*Figure 4C*). Based on this analysis, it appears that genomic variants of the infecting MTB strain may be markers for poor tuberculosis treatment outcomes.

## Discussion

To our knowledge, this is the first comprehensive evaluation of the contribution of bacterial genetic factors to poor treatment outcomes in drug-susceptible TB patients. Although our analysis showed that, as expected, patient characteristics of age, sex, and duration of diagnostic delay were associated with poor TB treatment outcomes, it also found fourteen bacterial genomic variants that were associated with an increased risk of poor treatment outcomes. The best clinical prediction model, with an AUC of 0.74, incorporates both host and bacterial risk factors. In a setting where the genomes of all MTB isolates are sequenced, these risk factors may provide a rationale for developing personalized approaches to tuberculosis treatment.

GWAS has yielded remarkable advances in the understanding of complex traits and has identified hundreds of genetic risk variants in humans (*Uffelmann et al., 2021*), including genetic polymorphisms associated with increased susceptibility to TB (*Quistrebert et al., 2021*; *Zheng et al., 2018*). With the increasing availability of bacterial WGS data, it is now possible to use GWAS to probe the relationship between pathogen genotypes and clinical disease phenotypes. GWAS has identified genetic determinants of MTB drug resistance (*Coll et al., 2018*; *Farhat et al., 2013*), TB transmission (*Sobkowiak et al., 2020*), virulence (*Genestet et al., 2022*) and host preference (*Luo et al., 2022*). In this study, GWAS was used to identify bacterial genetic variants associated with poor treatment outcomes in patients with drug-susceptible TB and assess their impact on outcomes.

While this study found mutations in the MTB genome that were associated with poor treatment outcomes, models that consider only bacterial factors were poor predictors of outcome, with an AUC of only 0.58. Furthermore, only 24.2% (22/91) of patients with poor outcomes carried at least one of the GWAS-identified fixed mutations (*Figure 2—source data 1*), and therefore these mutations played no role in the majority of poor outcomes. The 237 GWAS-identified unfixed mutations are diverse (*Figure 2—figure supplement 1*) and have a mutation frequency that is generally only 5%–10% (*Figure 2—figure supplement 3*). The low frequency of these diverse unfixed mutations suggests they may confer a relatively small selective advantage without any preferential mutation types or increasing mutation frequency. In addition, none of the GWAS-identified unfixed mutations were among the 14 GWAS-identified fixed mutations, suggesting the unfixed mutations merely reflect individual genetic features of the particular MTB isolate with no evidence of homoplastic fixation in the bacterial population.

There is evidence, although limited, that genes carrying the fourteen fixed GWAS-identified mutations play a role in the response to first-line drug and hypoxic stress. The *otsB1* gene harbors the mutation with the strongest association and encodes trehalose-6-phosphate phosphatase, which has been associated with rifampicin tolerance under hypoxic stress (*Jakkala and Ajitkumar, 2019*). The E2 component of pyruvate dehydrogenase, encoded by *dlaT*, is required for optimal MTB growth and resistance to reactive nitrogen intermediates (RNI) and immune killing by host cells (*Shi and Ehrt, 2006*; *Tian et al., 2005*). Most of the genes carrying the fourteen GWAS-identified fixed mutations were rarely reported in previous large-scale GWAS of drug-resistant MTB (*Coll et al., 2018*; *Farhat et al., 2013*; *Naz et al., 2023*), and none of the fourteen mutations were reported in these studies. However, two of the genes in which the mutations were found had been previously identified as potentially associated with first-line drug resistance (*Farhat et al., 2013*): CtpB, a probable cation-transporter P-type ATPase B; and MetA, a probable homoserine O-acetyltransferase. Our inference that the fourteen fixed mutations had only limited effects on treatment outcome would explain why:

they were not identified in previous studies; isolates from only 24.2% (22/91) of patients carried any of these 14 mutations; and none of the mutations were shared amongst all 22 patients.

The mutations may be epistatic adaptations suited to the genomic characteristics of each individual strain. The GWAS-identified mutations were also present in some patients with good outcomes but were less frequent than in patients with poor outcomes. Although patient information was unavailable concerning other known host risk factors such as adherence to the treatment regimen and comorbidities such as HIV co-infection, diabetes, smoking, and low BMI (*Kamara et al., 2022*; *Leung et al., 2015*; *Vernon et al., 2019*), our clinical prediction model still revealed that host factors are significantly more important determinants of poor outcomes than bacterial factors. Nevertheless, we believe that our study has shown that bacterial genomic factors can also contribute to poor outcomes.

Sample size and the classification of treatment outcomes are important challenges when exploring the association of bacterial factors with poor outcomes. Because standard first-line regimens cure at least 85% of drug-susceptible TB patients, this study had to pool data from three sites with a total of 3496 new cases to obtain 91 patients with poor outcomes. We excluded 300 patients who either died from non-TB causes, were lost to follow-up, or had unknown outcomes. We attempted to validate our findings with a dataset of 1397 new drug-susceptible TB cases from Malawi (*Guerra-Assunção et al., 2015*), but were unable to replicate the GWAS analysis because the only poor treatment outcome in the Malawi dataset was death, and it was impossible to distinguish the patients who succumbed to TB from those who died from non-TB causes (*Figure 2—figure supplement 4*).

In conclusion, we found that there are bacterial genomic variants that are significantly associated with poor treatment outcomes in drug-susceptible TB patients. Although host factors are clearly more important, the most accurate models for predicting poor treatment outcomes with drug-susceptible TB incorporate both host and bacterial risk factors. In the future, it may be possible to identify patients at high risk for treatment failure by analyzing the characteristics of both the host and the bacterial genome of the infecting strain. This could make it possible to identify patients requiring longer or individualized treatment regimens and thus improve cure rates for drug-susceptible TB.

## Materials and methods
### Selection of patients and samples

A strain database search was performed for TB patients treated during 2009–2020 at three study sites in Shanghai, Sichuan, and Heilongjiang, China. For each of the 4374 TB patients registered during this period, a pretreatment sputum sample was decontaminated and inoculated onto Löwenstein-Jensen (LJ) medium (Heilongjiang) or in liquid medium (Shanghai and Sichuan) and observed for 6–8 weeks. Culture-positive isolates were re-cultured on LJ medium for 3–4 weeks. Colonies were scraped from the surface of the LJ slopes and the DNA was isolated for WGS. WGS data and the patients' demographic and clinical features were obtained from a published study (*Li et al., 2022*). All new cases susceptible to first-line drugs (rifampicin, isoniazid, pyrazinamide, ethambutol) by genotypic drug-susceptibility testing (gDST), and whose records contained treatment outcomes, were selected for the study.

The WHO recommended treatment outcome definitions for TB are cured, treatment completed, treatment failed, died, lost to follow-up and not evaluated (*Linh et al., 2021*). Of these, patients who died were divided into deaths from TB and non-TB, and those not evaluated included cases transferred for treatment of multidrug-resistant tuberculosis (MDR-TB) and cases whose treatment outcome was unknown. For the current study, TB treatment outcomes were grouped into three categories: (1) good outcomes -- cured and treatment completed; (2) poor outcomes -- treatment failures, deaths from TB, transferred for MDR and relapse; and (3) other -- lost to follow-up, non-TB deaths and unknown outcome.

### SNPs calling, resistance prediction, and phylogenetic reconstruction

A previously described pipeline was used for calling single nucleotide polymorphisms (SNPs) (*Chen et al., 2021*). Briefly, the Sickle tool was used to trim WGS data to retain reads with a Phred base quality above 20 and a length greater than 30 nucleotides. Reads were mapped to the MTB H37Rv reference strain (GenBank AL123456) with bowtie2 (v2.2.9), and then SAMtools (v1.3.1) was used for SNP-calling with a mapping quality greater than 30. Varscan (v2.3.9) was used to identify fixed

(frequency, ≥75%) and unfixed (<75%) SNPs with at least 5 supporting reads and the strand bias filter option on. A previously validated pipeline was used to filter out false positives that may have arisen during the in vitro expansion of bacterial colonies or caused by PCR and sequencing errors (*Liu et al., 2022*). The drug-resistance profile and lineages were predicted from WGS data using SAM-TB (*Yang et al., 2022*). Phylogenetic trees were constructed using the maximum-likelihood method (RAxML-NG) (*Kozlov et al., 2019*) and visualized on the Interactive Tree of Life platform (https://itol.embl.de/).

## Estimates of nucleotide diversity and GWAS analyses

Nucleotide diversity ($\pi$) was estimated using the PoPoolation package (*Kofler et al., 2011*). Following O'Neill et al. (*O'Neill et al., 2015*), we randomly subsampled (n=10) read data from each sample to a uniform 50x coverage to limit the effects of differential coverage across samples. Using the subsampled data with uniform coverage, we then calculated nucleotide diversity in 100 kb sliding windows across the genome in 10 kb steps. GWAS analyses were performed using GEMMA software (v0.98.3) (*Zhou and Stephens, 2012*) to identify nonsynonymous variants associated with poor TB treatment outcomes. A linear mixed model was used to control for the confounding effects of MTB lineage, sublineage, and outbreak-based population structure (*Coll et al., 2018*). Host risk factors associated with poor treatment outcomes such as age, sex, and duration of diagnostic delay were included as covariates in the GWAS, and the significance threshold was adjusted with the Bonferroni correction.

## RNA-seq data collection and analysis

Raw RNA-Seq read data (GSE165581: INH, GSE166622: RIF, GSE118084: EMB, and GSE116353: hypoxia) from MTB laboratory strain H37Rv exposed to first-line drugs and hypoxic conditions was downloaded from the Gene Expression Omnibus (GEO) database (https://www.ncbi.nlm.nih.gov/geo/). Sequencing reads passing quality control was aligned to the MTB H37Rv reference strain using bowtie2. Unique reads were selected and sorted using SAMtools, then quantitated using htseq-count (v0.11.3). FPKM values calculated by DESeq2 (v1.26.0) were used as measures of gene expression, and genes with $|\log_2(\text{fold change})|\geq1$ and p-values <0.05 were considered differentially expressed.

## Statistical analysis

The t-test was used for comparing the mutations across gene functional categories in TubercuList (*Kapopoulou et al., 2011*) nucleotide diversity and the ratios of the six mutation types between TB patients with good and poor treatment outcomes. The mean was given plus or minus standard error (mean ± SE). The chi-square test was used to assess whether the distribution of MTB lineages differed between patients with different treatment outcomes. Factors associated with poor treatment outcomes were tested with logistic regression in univariate and multivariate analyses. Variables found to have a p<0.2 in the univariate analyses were included in the multivariate models. We constructed logistic regression models with the selected bacterial and host factors as predictors of TB treatment outcome, and the ROC curves of the prediction models were compared using DeLong's test.

## Acknowledgements

We thank Dr. Mia Crampin, Dr. Judith Glynn, and Ms Estelle McLean from the London School of Hygiene and Tropical Medicine for supplying or collating patient information on the Malawi dataset, and Dr. Qingyun Liu from Harvard T H Chan School of Public Health for the constructive comments provided on an earlier version of this paper. This study was supported by the National Natural Science Foundation of China (82272376 to QG), Shanghai Municipal Science and Technology Major Project (ZD2021CY001 to QG).

## Additional information

### Funding

| Funder | Grant reference number | Author |
|---|---|---|
| National Natural Science Foundation of China | 82272376 | Qian Gao |
| Shanghai Municipal Science and Technology Major Project | ZD2021CY001 | Qian Gao |

The funders had no role in study design, data collection and interpretation, or the decision to submit the work for publication.

### Author contributions

Yiwang Chen, Data curation, Software, Formal analysis, Investigation, Methodology, Writing – original draft, Writing – review and editing; Qi Jiang, Funding acquisition, Methodology, Writing – review and editing; Mijiti Peierdun, Methodology, Writing – review and editing; Howard E Takiff, Writing – review and editing; Qian Gao, Conceptualization, Resources, Supervision, Funding acquisition, Investigation, Project administration, Writing – review and editing

### Author ORCIDs

Qian Gao  http://orcid.org/0000-0002-8489-3672

### Decision letter and Author response

Decision letter https://doi.org/10.7554/eLife.84815.sa1
Author response https://doi.org/10.7554/eLife.84815.sa2

## Additional files

### Supplementary files

• MDAR checklist

### Data availability

Files containing sequencing reads were deposited in the National Institutes of Health Sequence Read Archive under BioProject PRJNA869190.

The following datasets were generated:

| Author(s) | Year | Dataset title | Dataset URL | Database and Identifier |
|---|---|---|---|---|
| Yi Chen, Jiang Q, Peierdun M | 2022 | The mutational signatures of poor treatment outcomes on the drug-susceptible Mycobacterium tuberculosis genome | https://www.ncbi.nlm.nih.gov/bioproject/?term=PRJNA869190 | NCBI BioProject, PRJNA869190 |
| Li M, Guo M, Peng Y | 2022 | High proportion of tuberculosis transmission among social contacts in rural China: a 12-year prospective population-based genomic epidemiological study | https://ngdc.cncb.ac.cn/bioproject/browse/PRJCA008815 | Genome Sequence Archive, PRJCA008815 |
| Li M, Guo M, Peng Y | 2022 | High proportion of tuberculosis transmission among social contacts in rural China: a 12-year prospective population-based genomic epidemiological study | https://ngdc.cncb.ac.cn/bioproject/browse/PRJCA008816 | Genome Sequence Archive, PRJCA008816 |

The following previously published datasets were used:

| Author(s) | Year | Dataset title | Dataset URL | Database and Identifier |
|-----------|------|---------------|-------------|-------------------------|
| Peterson EJ, Baliga NS | 2021 | Mycobacterium tuberculosis transcriptional response to Isoniazid | https://www.ncbi.nlm.nih.gov/geo/query/acc.cgi?acc=GSE165581 | NCBI Gene Expression Omnibus, GSE165581 |
| Peterson EJ, Baliga NS | 2021 | Mycobacterium tuberculosis transcriptional response to Rifampicin | https://www.ncbi.nlm.nih.gov/geo/query/acc.cgi?acc=GSE166622 | NCBI Gene Expression Omnibus, GSE166622 |
| Ma S, Lohmiller J, Sherman DR | 2018 | Transcriptional Response of Mycobacterium tuberculosis H37Rv to Ethambutol | https://www.ncbi.nlm.nih.gov/geo/query/acc.cgi?acc=GSE118084 | NCBI Gene Expression Omnibus, GSE118084 |
| Peterson EJ, Abidi AA, Baliga NS | 2019 | Intricate genetic program underlying hypoxia-induced dormancy in Mycobacterium tuberculosis revealed by high-resolution transcriptional time-course | https://www.ncbi.nlm.nih.gov/geo/query/acc.cgi?acc=GSE116353 | NCBI Gene Expression Omnibus, GSE116353 |
| Guerra-Assunção J, Crampin A, Houben R | 2015 | Large-scale whole genome sequencing of M. tuberculosis provides insights into transmission in a high prevalence area | https://www.ebi.ac.uk/ena/browser/view/ERS13471621 | European Nucleotide Archive, ERP000436 |
| Guerra-Assunção J, Crampin A, Houben R | 2015 | Large-scale whole genome sequencing of M. tuberculosis provides insights into transmission in a high prevalence area | https://www.ebi.ac.uk/ena/browser/view/ERP001072 | European Nucleotide Archive, ERP001072 |

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
