## [Editor Report]

In this useful study, a Genome Wide Association-type analysis is applied to clinical *Mycobacterium tuberculosis* isolates to discover genetic polymorphisms linked to poor tuberculosis outcomes. The evidence for the detected associations is still incomplete, as the corresponding polymorphisms are not adequate to power a prediction model for infection outcome, although key host factors – including patient age, sex, and duration of diagnostic delay (which have stronger predictive value) – appear to enhance predictive capacity. The work will be of interest to clinical microbiologists.

---

## [Decision Letter]

**Decision letter after peer review:**

Thank you for submitting your article "The mutational signatures of poor treatment outcomes on the drug-susceptible *Mycobacterium tuberculosis* genome" for consideration by *eLife*. Your article has been reviewed by 2 peer reviewers, and the evaluation has been overseen by a Reviewing Editor and Bavesh Kana as the Senior Editor. The following individual involved in the review of your submission has agreed to reveal their identity: Annelies Van Rie (Reviewer #1).

Essential Revisions:

1. The manuscript provides little to no detail about how the samples were obtained, other than the fact that they are "pre-treatment" samples. The authors must clarify if they are all sputum samples (as assumed) and if some/all represent induced sputa. Similarly, much greater detail is required about sample propagation, in particular, the possibility that samples underwent (multiple rounds of) in vitro culture to achieve sufficient biomass for whole-genome sequencing. If so, what growth media were used, how long were samples cultured, and how many generations does this equate to? Also, it is very important to know if all samples were treated identically.

2. Continuing from the above, it wasn't clear what was meant by some of the terminology used in the manuscript. Does the term "isolates" indicate that samples were plated to separate single colonies? Or are the "isolates" referred to throughout the manuscript actually populations of potentially different Mtb clones obtained – and propagated – as mixed samples? This information is critical given the potential that the identified polymorphisms – both fixed and, perhaps even more so, unfixed – might have arisen as a consequence of in vitro (laboratory) handling under standard aerobic conditions.

3. Causation is always difficult to establish unequivocally in this type of study. Ideally, the Mtb genotypes contained within samples obtained pre-treatment should be compared with samples obtained following treatment when the "poor" outcome was manifest – the expectation being that the polymorphisms identified should be evident (even dominant) at the later stage. In the absence of these data – which are difficult to obtain (even impossible, for example, where the "poor" outcome was death) – it would strengthen the perceived relevance of the identified mutations if the authors were able to provide any other evidence – perhaps from studies of drug-resistant Mtb isolates? – supporting their inferred role in undermining frontline treatment.

4. Related to the above, the authors make a valid point that their intention here was different from other studies which have deliberately utilized drug-resistant Mtb isolates to identify resistance-conferring and resistance-enabling mutations. Nevertheless, it would be very interesting to know if any of the mutations identified in those other studies were also picked up here – and, if not, why that might be the case.

5. The analyses presented in this study are heavily dependent on statistical methods; however, there is very little detail provided about the methodology applied. This omission detracts from the inferred outcomes-associated Mtb polymorphisms, especially given that the predominance of lineage 2 (and sub-lineage 2.3) risks a lineage-specific association, rather than a more generalizable pathogenicity phenotype.

6. It is also not clear that the statistical analyses have considered the fact that the "good" outcomes exceed the "poor" outcomes by a factor of ~34x. Similarly, the heavy skew in the "good" (3105) versus "poor" (91) samples (approximately 34x difference in sample size) raises the possibility that mutations identified in the "poor" category might be artificially over-represented. More clarity in detailing the statistical methods is required to allay any concerns about the identification of candidate polymorphisms.

7. Related to the above: The reference in the abstract to "3496 MTB strains" is deceptive: 300 "other" were excluded, and the remaining samples are split 3105:91. This means only 91 samples derive from "poor" outcomes, which is the focus of the work.

8. The authors note the importance of "host" factors, including age, sex, duration of diagnostic delay, etc. How were these factors separated (excluded) from the mycobacterium-associated factors in comparing genotypes from "good" versus "poor" outcomes?

9. It is not clear why SEM was performed. SEM is used to assess causal relationships between variables. This study is focused on the predictive power of the genomic mutations, on their own and adjusted for patient-related factors. SEM seems inappropriate here and can be removed.

10. In drug-susceptible TB, adherence to treatment is probably the most important determinant for failure and relapse. Were data on adherence included as a variable in the logistic regression model?

11. The following results are essential and should be included in the abstract: (i) Bacterial factors on their own had poor predictive power AUC of 0.58. The AUC for predicting poor treatment outcomes increased significantly (P = 0.01) from 0.70 to 0.74 when including both. (ii) Only 24.2% (22/91) of patients with poor outcomes carried at least one GWAS-identified fixed mutation.

12. In the conclusion, it is stated: "While we found a statistically significant effect of the MTB genomic determinants on treatment outcomes of drug-sensitive cases, the effect appears to be limited" This would be the most appropriate conclusion for the abstract.

*Reviewer #1 (Recommendations for the authors):*

Half of the poor treatment outcomes were relapses.

Q1: did you check if there were differences in the14 Mtb variants between the relapse cases and other reasons for poor treatment outcomes?

Q2: For the patients who relapsed, was the isolate at the time of relapse also sequenced? Di you find the same variants in the 13 genes of interest at the time of relapse in these samples?

It is not clear to me why a SEM was performed. SEM is used to assess causal relationships between variables. In your analysis, you are much more interested in the predictive power of the genomic mutations, on their own and adjusted for patient-related factors. I do not think SEM is appropriate here and suggest you remove these results.

In drug-susceptible TB, adherence to treatment is probably the most important determinant for failure and relapse. Did you have data on adherence was this included as a variable in the logistic regression model?

Comment: (1) Bacterial factors on their own had poor predictive power AUC of 0.58. The AUC for predicting poor treatment outcomes increased significantly (P = 0.01) from 0.70 to 0.74 when including both. (2) only 24.2% (22/91) of patients with poor outcomes carried at least one GWAS-identified fixed mutation. These results are essential and should be included in the abstract

In the conclusion, you state: "While we found a statistically significant effect of the MTB genomic determinants on treatment outcomes of drug-sensitive cases, the effect appears to be limited" This would be the most appropriate conclusion for the abstract.

*Reviewer #2 (Recommendations for the authors):*

There is a novelty in this study given the focus on drug-susceptible treatment outcomes, and the results presented support the value of this type of work. That said, there are some aspects of the manuscript that the authors might consider addressing to strengthen the claims made:

1. The manuscript provides little to no detail about how the samples were obtained, other than the fact that they are "pre-treatment" samples. The authors must clarify if they are all sputum samples (as assumed) and if some/all represent induced sputa. Similarly, much greater detail is required about sample propagation, in particular, the possibility that samples underwent (multiple rounds) of in vitro culture to achieve sufficient biomass for whole-genome sequencing. If so, what growth media were used, how long were samples cultured, and how many generations does this equate to? Also, it is very important to know if all samples were treated identically.

2. Continuing from the above, it wasn't clear what was meant by some of the terminology used in the manuscript. Does the term "isolates" indicate that samples were plated to separate single colonies? Or are the "isolates" referred to throughout the manuscript actually populations of potentially different Mtb clones obtained – and propagated – as a mixed sample? This information is critical given the potential that the identified polymorphisms – both fixed and, perhaps even more so, unfixed – might have arisen as a consequence of in vitro (laboratory) handling under standard aerobic conditions.

3. Causation is always difficult to establish unequivocally in this type of study. Ideally, the Mtb genotypes contained within samples obtained pre-treatment should be compared with samples obtained following treatment – when the "poor" outcome was manifest, the expectation being that the polymorphisms identified should be evident (even dominant) at the later stage. In the absence of these data – which I recognize are difficult to obtain (even impossible, for example where the "poor" outcome was death) – it would strengthen the perceived relevance of the identified mutations if the authors were able to provide any other evidence – perhaps from studies of drug-resistant Mtb isolates? – supporting their inferred role in undermining frontline treatment.

4. Related to the above, the authors make a valid point that their intention here was different from other studies which have deliberately utilized drug-resistant Mtb isolates to identify resistance-conferring and resistance-enabling mutations. Nevertheless, it would be very interesting to know if any of the mutations identified in those other studies were also picked up here – and, if not, why that might be the case.

5. The analyses presented in this study are heavily dependent on statistical methods. As for sample handling, however, there is very little detail provided about the methodology applied. This omission detracts from the interpretation, especially given that the predominance of lineage 2 (and sub-lineage 2.3) risks a lineage-specific association, rather than a more generalizable pathogenicity phenotype.

6. Related to the above: The reference in the abstract to "3496 MTB strains" is deceptive: 300 "other" were excluded, and the remaining samples are split 3105:91. This means only 91 samples derive from "poor" outcomes, which is the focus of the work.

7. Continuing from the above, it is not clear that the statistical analyses have taken into account the fact that the "good" outcomes exceed the "poor" outcomes by a factor of ~34x. Similarly, the heavy skew in the "good" (3105) versus "poor" (91) samples (approximately 34x difference in sample size) raises the possibility that mutations identified in the "poor" category might be artificially over-represented. More clarity in detailing the statistical methods is required to allay any concerns about the identification of candidate polymorphisms.

8. The authors note the importance of "host" factors, including age, sex, duration of diagnostic delay, etc. How were these factors separated (excluded) from the mycobacterium-associated factors in comparing genotypes from "good" versus "poor" outcomes?

---

## [Author Response]

Essential Revisions:1. The manuscript provides little to no detail about how the samples were obtained, other than the fact that they are "pre-treatment" samples. The authors must clarify if they are all sputum samples (as assumed) and if some/all represent induced sputa. Similarly, much greater detail is required about sample propagation, in particular, the possibility that samples underwent (multiple rounds of) in vitro culture to achieve sufficient biomass for whole-genome sequencing. If so, what growth media were used, how long were samples cultured, and how many generations does this equate to? Also, it is very important to know if all samples were treated identically.

Thank you for pointing out this omission. Because the WGS data of the strains used in this study, along with details on their origins and culture, are described in detail our recently published paper ^[1]^, this information was not repeated in the current manuscript. Detailed information about sample propagation is as follows:

1. Sputum specimens were collected when the patients presented for tuberculosis diagnosis, before starting treatment.

2. Each sputum sample was decontaminated and cultured on solid Löwenstein-Jensen (L-J) medium (in Heilongjiang) or in liquid medium (in Shanghai and Sichuan) for 6-8 weeks before a negative result is reported.

Culture-positive isolates were re-cultured on solid L-J medium for 3-4 weeks, and then the colonies were scraped from the slope surface and the DNA was extracted using the CTAB method ^[2]^ and used for sequencing.

We have added this information to the methods section (lines 256-261): “For each of the 4374 TB patients registered during this period, a pretreatment sputum sample was decontaminated and inoculated onto Löwenstein-Jensen (LJ) medium (Heilongjiang) or in liquid medium (Shanghai and Sichuan) and observed for 6-8 weeks. Culture-positive isolates were re-cultured on LJ medium for 3-4 weeks. Colonies were scraped from the surface of the LJ slopes and the DNA was isolated for WGS”.

2. Continuing from the above, it wasn't clear what was meant by some of the terminology used in the manuscript. Does the term "isolates" indicate that samples were plated to separate single colonies? Or are the "isolates" referred to throughout the manuscript actually populations of potentially different Mtb clones obtained – and propagated – as mixed samples? This information is critical given the potential that the identified polymorphisms – both fixed and, perhaps even more so, unfixed – might have arisen as a consequence of in vitro (laboratory) handling under standard aerobic conditions.

Apologies for the confusion and thank you for the opportunity to clarify our work. As stated above, the “isolates” for WGS referred to the populations of colonies re-cultured from frozen stocks. We agree with the reviewers that new, false positive mutations could be generated during in vitro expansion of bacterial colonies or by PCR and sequencing errors. To filter out false positives, we used a previously validated pipeline developed in our lab. To confirm the reliability of our method, we selected three single colonies from different frozen stocks, grew them in L-J medium and subjected them to WGS using the same strategy employed in this study.

For single colonies, the unfixed mutations detected in the raw WGS data were considered to be false positive mutations arising during in vitro culture or the result of sequencing errors (Author response image 1). Most false positives (~99.6%) can be excluded by eliminating mutations with allele frequency < 5% or supported by fewer than five reads. The remaining false positives, such as mutations with more than 50% of supporting alleles enriched in the terminal region of sequencing reads, can be removed with our previously described pipeline trimming ^[3]^. To make it clearer we have added this additional information to the methods section (lines 283-285): “A previously validated pipeline was used to filter out false positives that may have arisen during the in vitro expansion of bacterial colonies or caused by PCR and sequencing errors (Liu et al., 2022)”.

**Author response image 1. sa2fig1:** Schematics of false positive filter in three single colonies. The density plot above the scatter plot shows the distribution of mutation depth while the plot to the right of the scatter plot shows the distribution of mutation frequency. The first row shows the SNP calling results from the raw data, in which there were many false positive mutations (FPMs). The second row shows the results after most FPMs were filtered out, leaving only those SNPs with frequency greater than 5% (horizontal yellow dashed line) and sequencing depth greater than 5 (vertical red dashed line). The third row shows the results after the remaining FPMs were filtered out with our validated pipeline.

3. Causation is always difficult to establish unequivocally in this type of study. Ideally, the Mtb genotypes contained within samples obtained pre-treatment should be compared with samples obtained following treatment when the "poor" outcome was manifest – the expectation being that the polymorphisms identified should be evident (even dominant) at the later stage. In the absence of these data – which are difficult to obtain (even impossible, for example, where the "poor" outcome was death) – it would strengthen the perceived relevance of the identified mutations if the authors were able to provide any other evidence – perhaps from studies of drug-resistant Mtb isolates? – supporting their inferred role in undermining frontline treatment.4. Related to the above, the authors make a valid point that their intention here was different from other studies which have deliberately utilized drug-resistant Mtb isolates to identify resistance-conferring and resistance-enabling mutations. Nevertheless, it would be very interesting to know if any of the mutations identified in those other studies were also picked up here – and, if not, why that might be the case.

Thank you for these insightful questions. We agree with the reviewer that comparison of pre- and post-treatment samples can better illustrate the causal relationship between the mutations we identified and poor treatment outcomes. However, post-treatment samples could only be obtained from the 47 relapse cases at the time of relapse. GWAS-identified fixed mutations in the relapse samples were only detected in the isolates from the 13 patients whose first samples also contained GWAS-identified mutations. Then, as suggested by the reviewer, we sought to explain the correlation between these mutations and the poor treatment outcomes in terms of their association with drug resistance.

We looked for mention of the mutations we identified in published studies of drug-resistant Mtb isolates ^[5-7]^. None of the 14 mutations we identified were reported in these studies, but two of the genes in which the mutations occurred (*ctpB* and *metA*) had been described as potentially associated with first-line drug resistance ^[5]^. We postulate two possible reasons that could explain why the mutations we found were not identified previously: (1) Differences in the genotype composition of the datasets may have led to inconsistent results. The datasets of previous studies were predominantly comprised of L4 strains, while ours included mainly L2 strains. (2) The 14 mutations we identified may have had only limited effects on treatment outcome and therefore did not display the convergent evolution seen with the drug-resistant mutations, especially those most frequent in clinical strains, such as *rpoB* S450L or *katG* S315T.

To test the first hypothesis, we used WGS data to select the drug-resistant strains collected during the same period at the three study sites and performed GWAS to identify mutations associated with resistance, but this analysis did not detect any of the 14 mutations associated with poor outcomes of drug-susceptible tuberculosis. This result was not surprising, because the strains we included in the study were pretreatment drug-susceptible isolates without prior drug exposure that would have selected for resistance associated mutations. We therefore favor the second explanation, i.e., the limited impact of the 14 mutations. Bacterial factors alone had low power for predicting poor outcomes, with an AUC of just 0.58. Only 24.2% (22/91) of patients with poor outcomes carried at least one GWAS-identified fixed mutation.

In conclusion, we believe our study demonstrates that there are genetic variants associated with poor outcomes in drug-susceptible Mtb, but their role in causing poor outcomes is limited. These results emphasize that future studies on bacterial genetic factors associated with poor treatment outcomes should focus on drug resistance-associated variants.

5. The analyses presented in this study are heavily dependent on statistical methods; however, there is very little detail provided about the methodology applied. This omission detracts from the inferred outcomes-associated Mtb polymorphisms, especially given that the predominance of lineage 2 (and sub-lineage 2.3) risks a lineage-specific association, rather than a more generalizable pathogenicity phenotype.

Thank you for pointing this out. We describe the statistical methods used for GWAS in the methods section (lines 299-300). Population structure has an impact on GWAS analyses, so after consulting previous GWAS strategies we used a linear mixed model to control for the confounding effects of MTB lineage. The distribution of most (85.7%, 12/14) of the inferred outcome-associated mutations across the lineages is not significantly different (Author response table 1). Although two mutations (*Rv1747* T191A/*cobN* A751V) were significantly more prevalent in L4 strains than in L2 strains, these two mutations were not specific to L4 strains and evolved convergently in strains of different lineages.

**Author response table 1. sa2table1:** Comparison of the ratio of strains carrying the GWAS-identified fixed mutation in different lineages.

		L2	L4	P-value				L2	L4	P-value
*Rv0051* Q149H	Yes	5	5	0.13^F^		*ctpB*E345K	Yes	27	15	0.14
	No	2368	818				No	2346	808	
										
*Rv0260c*T72I	Yes	41	22	0.09		*Rv0648*P454S	Yes	87	42	0.07
	No	2332	801				No	2286	781	
										
*Rv1248c**1232S	Yes	87	43	0.05		*Rv1747*T191A	Yes	94	66	<0.001
	No	2286	780				No	2279	757	
										
*otsB1*G559D	Yes	14	9	0.14		*cobN*A751V	Yes	98	60	<0.001
	No	2359	814				No	2275	763	
										
*Rv2164c*D233G	Yes	96	46	0.06		*dlaT*V55A	Yes	94	50	0.01
	No	2277	777				No	2279	773	
										
*Rv3168*E308*	Yes	17	9	0.30		*metA*G146D	Yes	106	50	0.07
	No	2356	814				No	2267	773	
										
*metA* E149G	Yes	104	56	0.01		*papA1*I497T	Yes	5	1	1^F^
	No	2273	763				No	2368	822	

Significant: p < 0.004 (0.05/14) after Bonferroni correction; F: Fisher exact test.

6. It is also not clear that the statistical analyses have considered the fact that the "good" outcomes exceed the "poor" outcomes by a factor of ~34x. Similarly, the heavy skew in the "good" (3105) versus "poor" (91) samples (approximately 34x difference in sample size) raises the possibility that mutations identified in the "poor" category might be artificially over-represented. More clarity in detailing the statistical methods is required to allay any concerns about the identification of candidate polymorphisms.

Thank you for this question. We also considered the effect of the ratio of controls to cases on the statistical power. When cases and controls are both freely available, selecting equal numbers will make a study most efficient. Because of the limited number of cases in this study, i.e., drug susceptible TB patients with poor outcomes, we attempted to augment the statistical confidence by increasing the ratio of controls to cases. We are aware that the gains in statistical power diminish as the ratio of controls to cases increases, and at ratios greater than 4:1 the additional statistical power is minimal. However, if the data on controls is easily obtained there is no reason to limit the number of controls ^[8]^. As our study design was based on a previously collected population-based dataset, there was no basis for limiting the ratio of controls to cases.

7. Related to the above: The reference in the abstract to "3496 MTB strains" is deceptive: 300 "other" were excluded, and the remaining samples are split 3105:91. This means only 91 samples derive from "poor" outcomes, which is the focus of the work.

Thank you for your advice. As suggested, a clear description of the included samples has been added to the abstract section (lines 24-26): *“*We analyzed whole-genome sequencing (WGS) data of MTB strains from 3196 patients, including 3105 patients with good and 91 patients with poor treatment outcomes, and linked genomes to patient epidemiological data*”.*

8. The authors note the importance of "host" factors, including age, sex, duration of diagnostic delay, etc. How were these factors separated (excluded) from the mycobacterium-associated factors in comparing genotypes from "good" versus "poor" outcomes?

Thank you for pointing this out, and apologies for not describing this in detail. We conducted multivariate logistic regression analyses based on host factors before performing GWAS analyses, and then used age, sex and duration of diagnostic delay as covariates for GWAS analyses. The results of our logistic regression presented in the manuscript (Figure 4A) indicate that GWAS-identified mutations are independently associated with poor treatment outcomes after correcting for these host factors. We have added this information to the methods section (lines 299-300): “host risk factors associated with poor treatment outcomes such as age, sex and duration of diagnostic delay were included as covariates in the GWAS”.

9. It is not clear why SEM was performed. SEM is used to assess causal relationships between variables. This study is focused on the predictive power of the genomic mutations, on their own and adjusted for patient-related factors. SEM seems inappropriate here and can be removed.

Thank you for your advice. We used SEM to try to represent the degree of influence of bacterial and host factors on treatment outcome through path coefficients ^[9]^, but this part of the result is actually illustrated by the ROC curve. We agree with the reviewer that SEM may not be essential for our results and have removed it in the revised manuscript.

10. In drug-susceptible TB, adherence to treatment is probably the most important determinant for failure and relapse. Were data on adherence included as a variable in the logistic regression model?

Thank you for this question. We did not include adherence as a variable in the logistic regression model because all included patients completed the standard treatment course for new cases with drug-susceptible TB. They all had full records of medication taken and treatment outcomes according to the national TB programme guidelines ^[10]^. Those patients whose treatment was interrupted and had no clear treatment outcome were excluded from our initial analysis, so it was not possible to consider treatment interruption as a candidate risk factor for poor outcomes.

11. The following results are essential and should be included in the abstract: (i) Bacterial factors on their own had poor predictive power AUC of 0.58. The AUC for predicting poor treatment outcomes increased significantly (P = 0.01) from 0.70 to 0.74 when including both. (ii) Only 24.2% (22/91) of patients with poor outcomes carried at least one GWAS-identified fixed mutation.

Thank you for this advice. We have added these two points to the abstract (lines 30-32 and 36-39).

12. In the conclusion, it is stated: "While we found a statistically significant effect of the MTB genomic determinants on treatment outcomes of drug-sensitive cases, the effect appears to be limited" This would be the most appropriate conclusion for the abstract.

Thank you again for your advice. We agree that it is important to highlight the “limited” impact of MTB genomic determinants and have added this to the conclusions in the abstract (lines 39-41).

Reviewer #1 (Recommendations for the authors): Half of the poor treatment outcomes were relapses. Q1: did you check if there were differences in the14 Mtb variants between the relapse cases and other reasons for poor treatment outcomes?

Thank you for this question. The ratio of strains from relapse cases that carried at least one GWAS-identified fixed mutation is higher than that from all other patients (27.7% vs 7.7%, Fishers exact test, p < 0.001, Author response table 2), but is not significantly different from the ratio in patients with other poor treatment outcomes (27.7% vs 20.5%, chi-square test, p = 0.422, Author response table 3).

**Author response table 2. sa2table2:** Comparison of the ratios of strains carrying at least one GWAS-identified fixed mutation from relapse cases with strains from all other patients.

		GWAS-identified mutations	P-value	
		Yes	No	
Relapse	Yes	13	34	P < 0.001
	No	241	2908	

**Author response table 3. sa2table3:** Comparison of the ratio of strains carrying at least one GWAS-identified fixed mutation from the relapse cases with strains from patients with other poor treatment outcomes.

		GWAS-identified mutations	P-value	
		Yes	No	
Relapse	Yes	13	34	P = 0.422
	No	9	35	

Q2: For the patients who relapsed, was the isolate at the time of relapse also sequenced? Di you find the same variants in the 13 genes of interest at the time of relapse in these samples?

Yes, we sequenced the isolates at the time of relapse from all 47 relapse cases, and found that the 14 GWAS-identified fixed mutations were only detected in relapse isolates from the 13 patients whose first samples also contained GWAS-identified mutations. None of the 14 mutations we identified were detected in the samples from other relapsed patients obtained at the time of relapse.

It is not clear to me why a SEM was performed. SEM is used to assess causal relationships between variables. In your analysis, you are much more interested in the predictive power of the genomic mutations, on their own and adjusted for patient-related factors. I do not think SEM is appropriate here and suggest you remove these results.

Thank you for this question, the SEM has been removed from the revised manuscript. For details, please see our response, above, to the essential revisions (9).

In drug-susceptible TB, adherence to treatment is probably the most important determinant for failure and relapse. Did you have data on adherence was this included as a variable in the logistic regression model?

Thank you for this question, we did not include adherence as a variable in the logistic regression analysis because those patients whose treatment was interrupted with no clear outcome were excluded from our initial analysis. For details, please see our response, above, to the essential revisions (10).

Comment: (1) Bacterial factors on their own had poor predictive power AUC of 0.58. The AUC for predicting poor treatment outcomes increased significantly (P = 0.01) from 0.70 to 0.74 when including both. (2) only 24.2% (22/91) of patients with poor outcomes carried at least one GWAS-identified fixed mutation. These results are essential and should be included in the abstract

Thank you for this suggestion, these changes have been made in the abstract of the revised version of the manuscript.

In the conclusion, you state: "While we found a statistically significant effect of the MTB genomic determinants on treatment outcomes of drug-sensitive cases, the effect appears to be limited" This would be the most appropriate conclusion for the abstract.

Thank you for this suggestion. The abstract has been changed, accordingly, in the new version of the manuscript.

Reviewer #2 (Recommendations for the authors):There is a novelty in this study given the focus on drug-susceptible treatment outcomes, and the results presented support the value of this type of work. That said, there are some aspects of the manuscript that the authors might consider addressing to strengthen the claims made:1. The manuscript provides little to no detail about how the samples were obtained, other than the fact that they are "pre-treatment" samples. The authors must clarify if they are all sputum samples (as assumed) and if some/all represent induced sputa. Similarly, much greater detail is required about sample propagation, in particular, the possibility that samples underwent (multiple rounds) of in vitro culture to achieve sufficient biomass for whole-genome sequencing. If so, what growth media were used, how long were samples cultured, and how many generations does this equate to? Also, it is very important to know if all samples were treated identically.2. Continuing from the above, it wasn't clear what was meant by some of the terminology used in the manuscript. Does the term "isolates" indicate that samples were plated to separate single colonies? Or are the "isolates" referred to throughout the manuscript actually populations of potentially different Mtb clones obtained – and propagated – as a mixed sample? This information is critical given the potential that the identified polymorphisms – both fixed and, perhaps even more so, unfixed – might have arisen as a consequence of in vitro (laboratory) handling under standard aerobic conditions.3. Causation is always difficult to establish unequivocally in this type of study. Ideally, the Mtb genotypes contained within samples obtained pre-treatment should be compared with samples obtained following treatment – when the "poor" outcome was manifest, the expectation being that the polymorphisms identified should be evident (even dominant) at the later stage. In the absence of these data – which I recognize are difficult to obtain (even impossible, for example where the "poor" outcome was death) – it would strengthen the perceived relevance of the identified mutations if the authors were able to provide any other evidence – perhaps from studies of drug-resistant Mtb isolates? – supporting their inferred role in undermining frontline treatment.4. Related to the above, the authors make a valid point that their intention here was different from other studies which have deliberately utilized drug-resistant Mtb isolates to identify resistance-conferring and resistance-enabling mutations. Nevertheless, it would be very interesting to know if any of the mutations identified in those other studies were also picked up here – and, if not, why that might be the case.5. The analyses presented in this study are heavily dependent on statistical methods. As for sample handling, however, there is very little detail provided about the methodology applied. This omission detracts from the interpretation, especially given that the predominance of lineage 2 (and sub-lineage 2.3) risks a lineage-specific association, rather than a more generalizable pathogenicity phenotype.6. Related to the above: The reference in the abstract to "3496 MTB strains" is deceptive: 300 "other" were excluded, and the remaining samples are split 3105:91. This means only 91 samples derive from "poor" outcomes, which is the focus of the work.7. Continuing from the above, it is not clear that the statistical analyses have taken into account the fact that the "good" outcomes exceed the "poor" outcomes by a factor of ~34x. Similarly, the heavy skew in the "good" (3105) versus "poor" (91) samples (approximately 34x difference in sample size) raises the possibility that mutations identified in the "poor" category might be artificially over-represented. More clarity in detailing the statistical methods is required to allay any concerns about the identification of candidate polymorphisms.8. The authors note the importance of "host" factors, including age, sex, duration of diagnostic delay, etc. How were these factors separated (excluded) from the mycobacterium-associated factors in comparing genotypes from "good" versus "poor" outcomes?

Each of these questions has been carefully answered in our responses to the essential revisions, above.